# MOF-Derived Nanoporous Carbon Incorporated in the Cation Exchange Membrane for Gradient Power Generation

**DOI:** 10.3390/membranes12030322

**Published:** 2022-03-14

**Authors:** Xia Sun, Ying Liu, Ruibo Xu, Yongsheng Chen

**Affiliations:** 1School of Environmental and Chemical Engineering, Jiangsu Ocean University, Lianyungang 222005, China; daying1223@163.com; 2Jiangsu Marine Resources Development Research Institute, Lianyungang 222005, China; 3School of Pharmacy, Jiangsu Ocean University, Lianyungang 222005, China; xuribo9125@163.com; 4Georgia Institute of Technology, School of Civil and Environmental Engineering, Atlanta, GA 30332, USA

**Keywords:** metal-organic framework (MOF), porous carbon, cation exchange membrane, reverse electrodialysis, salinity gradient power

## Abstract

Ion exchange membranes (IEMs), as a part of the reverse electrodialysis (RED) system, play an important role in salinity gradient power (SGP) generation. Structure optimization of IEMs is critical to increase the power production by RED. In this work, metal organic framework (MOF)-derived nanoporous carbons (hollow zeolitic imidazolate framework (ZIF)-derived nanoporous carbons, HZCs) were incorporated in a sulfonated poly (2, 6-dimethyl-1,4-phenylene oxide) (sPPO) membrane to prepare an organic−inorganic nanocomposite cation exchange membrane (CEM). Physicochemical properties, electrochemical properties, and power generation of the synthesized nanocomposite membranes with different HZCs loading were characterized. The results show that the incorporated HZCs could tailor the microstructure of the membrane matrix, providing a superior performance of the nanocomposite membrane. With a HZCs loading of 1.0 wt.%, the nanocomposite membrane possessed the highest permselectivity of 77.61% and the lowest area resistance of 0.42 Ω·cm^2^, along with a super gross power density of 0.45 W/m^2^, which was 87.5% (about 1.87 times) higher than that of the blank sPPO membrane. Therefore, incorporating of an appropriate amount of HZCs in the ion-exchange membrane can improve the performance of the membrane, providing a promising method to increase the power generation of the RED system.

## 1. Introduction

With energy shortage being such a serious worldwide problem, renewable and sustainable energy resources continue to attract more attention. The ocean is a huge energy reservoir, which can provide energy flow as an energy source [1]. Energy, obtained by the reversible mixing of two streams with different salt concentrations, is called salinity gradient power [2]. Seawater and fresh water can produce 0.8 kilowatts of power per cubic meter, which is equal to the amount of energy generated by a 280-m high dam [3]. Therefore, research on energy conversion technology has become vital to our planet’s need for cleaner energy that can reduce the carbon footprint. Membrane-based technology is believed to be a promising technology to produce SGP, such as reverse electrodialysis (RED) [4] and pressure-retarded osmosis (PRO) [5].

RED is an attracting technology that does not pollute the environment in the process of power generation. Devices can be set up wherever two solutions with different salinity are mixed, generating electricity using ion exchange membranes (IEMs) [6]. A typical RED system consists of three parts: the membrane stack, separator, and electrode system [7,8]. As the core part of RED, IEMs, including cation exchange membranes (CEMs) and anion exchange membranes (AEMs), which are polymer electrolytes containing ions, are separated by woven fabric spacers and are arranged alternately to form small compartments with fixed distances. The salt solution and dilute salt solution pass alternately into the compartments, in which cations diffuse to the cathode through CEMs and anions diffuse to the anode through AEMs. The redox reaction takes place at two electrodes to convert ion current into electric current. Therefore, the microstructures and properties of IEMs have a great impact on the RED system, including the surface morphology, thickness, swelling degree (SD), ion exchange capacity (IEC), permselectivity, and resistance of membranes [9]. Several results indicate that an ideal IEM for RED should have a low ionic resistance, high IEC, and high permselectivity [10]. However, it is still a challenge to prepare a membrane with these properties because they can counteract and have an impact on performance of the membrane [11,12].

Traditional organic IEMs have the advantage of good flexibility and low density, but they also have poor stability. Inorganic fillers possess a high specific surface area and strong hydrophilicity on merit [2]. Blending an organic polymer matrix with inorganic fillers to form an organic−inorganic nanocomposite membrane can remarkably improve the IEMs properties, for example through increased porosity and permselectivity, improved IEC and electrical conductivity, and a reduction in resistance of the membrane, which can effectively improve the performance of the RED system [12]. Therefore, by combining the advantages of organic and inorganic materials, the organic−inorganic nanocomposite membrane has become a research hotspot [13].

Currently, a series of nanofillers, such as poly (diallyldimethylammonium chloride) (PDDA) [7], poly (vinyl alcohol) (PVA) [11], SiO_2_–SO_3_H [12], oxidized multi-walled carbon nanotubes [14], and Fe_2_O_3_–SO_4_^2−^ [2,15], have been involved in the IEMs matrix to improve the performance of IEMs, compared with their employed commercial IEMs. However, compared with the blank membranes used in the stacks, such nanomaterials did not remarkedly increase the power generation of the RED system, except for the nanomaterial of PDDA (shown in Table 1). As a new kind of crystalline porous material, metal-organic framework (MOF), composed of metal ions and organic ligands, has attracted extensive attention and has been widely studied in many fields, such as in gas storage and separation [16], as well as catalysis [17]. MOF is also an ideal template and precursor for the preparation of nanoporous carbon, due to its large specific surface areas, diverse structure, and functional well-defined pores [18]. Shen et al. [19] incorporated hollow ZIFs-derived nanoporous carbons (HZCs) into a graphite plate for capacitive deionization applications. The synthesized hollow ZIFs-derived nanoporous carbons significantly improve the performance of capacitive deionization. Huang et al. [20] employed sulfonated spindle-like carbon derived from MOF as a filler for the sulfonated poly (ether ether ketone) (SPEEK) membrane. The obtained MOF-C-SO_3_H@SPEEK membrane has shown improved properties as a polyelectrolyte multilayer (PEM)-based membrane for direct methanol fuel cells (DMFCs). However, there is no report on whether MOF-derived nanoporous carbon is used as a filler to improve the performance of IEMs in a RED system.

In this study, we developed a kind of organic–inorganic nanocomposite CEM through the incorporation of sulfonated poly (2,6-dimethyl-1,4-phenylene oxide) (sPPO) with MOF-derived nanoporous carbon. PPO is a commonly used polymer in membranes due to its good chemical, thermal, and hydrolytic stability; high glass transition temperature (t_g_ = 210 °C); and low cost [2,21]. In order to make PPO polymer more ion-exchangeable and highly hydrophilic, sulfonate groups (–SO_3_^−^) are introduced into the polymer chain via a sulfonating agent such as chlorosulfonic acid. MOF-derived nanoporous carbon, used as filler to be incorporated with organic polymers, are expected to tailor the microstructure so as to improve the physical and electrochemical properties of the nanocomposite membrane, as well as the power generation. The zeolite imidazolate framework-8 (ZIF-8), made of zinc ions and imidazolate ligands [19], was used as a precursor to prepare hollow ZIF-8 carbonized nanoparticles (HZCs). Here, the structure and morphology of the nanocomposite membranes were characterized, and the physicochemical and electrochemical properties, as well as the power production of the synthesized membranes were evaluated.

## 2. Materials and Experiments

### 2.1. Materials

Poly (2,6-dimethyl-1,4-phenylene oxide) (PPO) (Aldrich, analytical standard), chloroform, hydrochloric acid (HCl), sodium chloride (NaCl), and sodium hydroxide (NaOH) were all purchased from Sinopharm Group Chemical Reagent. Chlorosulfonic acid, dimethyl sulfoxide (DMSO), 2-methyimidazole (2-MeIM), tannic acid (TA), potassium ferricyanide (K_3_Fe(CN)_6_), potassium ferrocyanide (K_4_Fe(CN)_6_), and cetyltrimethylammonium bromide (CTAB) were purchased from Aladdin Industrial Corporation. All of the chemicals were analytical pure grade.

### 2.2. Preparation of CEMs

#### 2.2.1. Synthesis of sPPO

sPPO was synthesized by a previously reported method [2]. First, 6% PPO was dissolved in chloroform and stirred with a magnetic stirrer for 30 min until completely dissolved. Then, 8% chlorosulfonic acid was added dropwise into the mixed solution with another 30 min of stirring until sulfonated PPO precipitated. The resultant polymer was washed to be neutral, followed by filtration and was dissolved in methanol to form a 64 wt.% solution. Subsequently, the polymer solution was poured onto the surface of a sheet of smooth glass and dried at room temperature. sPPO was obtained after the dried polymer was broken up into pieces, washed with deionized (DI) water, and dried at room temperature. 

#### 2.2.2. Preparation of HZCs

ZIF-8 cubic nanocrystals and hollow ZIF-8 (HZIF-8) nanocubes were prepared by a typical tanning etching method [19]. Briefly, 10.8 g 2-MeIM and 0.7 g Zn(NO_3_)_2_·6H_2_O were separately dissolved into 100 mL of DI water. Then, 4 mL 0.01 M of CTAB was added into the 2-MeIM aqueous solution with stirring for 10 min. Then, the above solution was mixed with the Zn(NO_3_)_2_·6H_2_O aqueous solution. The resultant solution was stirred continuously for another 5 min. A white and milky solution was observed. After precipitation for more than 3 h, white ZIF-8 powder was obtained after the white precipitate was washed with DI water and anhydrous ethanol, followed by being dried overnight in a vacuum.

Then, 100 mg ZIF-8 was dissolved into 25 mL of anhydrous ethanol and was ultrasonicated to form a disperse solution. An appropriate amount of TA was dissolved in 25 mL deionized water to form a 20 g·L^−1^ TA solution. The as-prepared ZIF-8 solution and TA solution were mixed with continuous stirring and then aged at room temperature for another 5 min. Subsequently, the above solution was centrifuged and washed with deionized water, followed by anhydrous ethanol. Then, the off-white powder of HZIF-8 was obtained after being dried under a vacuum. HZCs were harvested by the calcination of HZIF-8 at 900 °C for 3 h under an N_2_ atmosphere. 

#### 2.2.3. Synthesis of the Nanocomposite CEM

A solvent evaporation technique was adopted to prepare the CEM [2]. The sPPO polymer was dissolved in DMSO to form a 19 wt.% solution. Then, various amounts of HZCs (0–1.5 wt.%) were added into the mixture with stirring for 24 h at 60 °C, and were sonicating for 10 min. After dispersion, the solution was cast on a glass substrate with a doctor blade to harvest the membrane, with a thickness of 20 μm. The membrane was dried in a vacuum before being immersed into a 50 °C deionized water bath. After 15–20 min, the membrane was peeled off from the glass substrate, treated with 1 M HCl and washed with DI water. The obtained nanocomposite CEM was immersed in a 0.5 M NaCl solution before application. 

### 2.3. Characterization

The morphology of the MOFs was analyzed by an FEI Quanta 250 FEG field emission scanning electron microscope (SEM) (FEI Co. Hillsboro, OR, USA) and the Philips TECNAI 12 transmission electron microscope (TEM) (Phillips Healthcare, Amsterdam, The Netherlands) under the voltage of 120 kV. X-ray diffraction (XRD) was operated on an advance powder diffraction system (PANALYTICAL, Almelo, The Netherlands). Fourier transform infrared (FTIR) spectra were obtained via a Nicolet-iS 10 spectrometers (Thermo Fisher Scientific, Ashville, NC, USA). The morphology of the membrane was obtained by a JSM-6390LA scanning electron microscope (Tokyo, Japan). In order to get smooth cross sections, the membrane was immersed into liquid nitrogen and then cut into small pieces. The thermal properties of the samples were found using a thermogravimetric analyzer (TGA) (NETZSCH STA 449F3, Selbu, Germany). Samples were heated from 50 to 800 °C at a heating rate of 10 °C/min under a nitrogen flow. The mechanical properties (tensile strength and elongation at break) were measured using CMT6103 MTS at 25 °C.

### 2.4. Physicochemical and Electrochemical Properties of Membrane

#### 2.4.1. Swelling Degree (SD)

SD is one of the key properties of IEMs, which is the amount of water in the per unit weight dry membrane. The as-developed membrane was immersed into DI water for about one day. After the surface water was removed, the wetted membrane was weighted and then dried at 50 °C to a constant weight. The SD test results presented in this paper calculated the swelling percentage of the membrane as follows: (1)$$\mathrm{SD}=\frac{W_{wet}-W_{dry}}{W_{dry}}\times 100\%$$
where *W_wet_* (g) and *W_dry_* (g) are the weight of the wet and dry membrane, respectively.

#### 2.4.2. Ion-Exchange Capacity (IEC)

A classical titration method was used to measure the IEC of the IEMs. The membrane was first dried at 50 °C to obtain a dry weight, and was then immersed in 1M HCl for about 15 h before rinsing with deionized water. The rinsed membrane was immersed in 1 M NaCl solution with another 6 h for the exchange of sodium ions with the equilibrated hydrogen ions. The NaOH solution (0.1 M) was adopted to titrate the replaced hydrogen ions. The IEC values were calculated as follows: (2)$$\mathrm{IEC}=\frac{C_{NaOH}\times V_{NaOH}}{W_{dry}}$$
where *C_NaOH_* is the concentration of NaOH solution (mol·L^−1^), *V_NaOH_* is the volume of NaOH solution (L) used, and *W_dry_* is the dry weight (g) of the membrane.

#### 2.4.3. Permselectivity

A static membrane potential measurement was used to test the permselectivity of the membrane [12], which was carried out in a facility with two cells separated by the tested membrane with an effective area of 1.78 cm^2^. The NaCl solution (0.1 M and 0.5 M) was filled in the two cells, respectively. The membrane samples were immersed in 0.1 M NaCl solution for 24 h before measurement. The potential difference across the membrane was measured using UT61E palm-size mini digital multimeter (Huazheng Electric Manufacturing Co., Ltd., Baoding, China) with two CH instruments, after the working system was stable for about 30 min. The permselectivity of the tested membrane was obtained as follows: (3)$$\alpha \left(\%\right)=\frac{\Delta \mathrm{measured}}{\Delta \mathrm{theoretical}}\times 100$$
where α is the membrane permselectivity (%), Δmeasured is the measured membrane potential (V), and Δtheoretical is the theoretical membrane potential (V). Δtheoretical is 0.0379 V calculated from the Nernst equation, which refers to 100% permselectivity of the membrane [22].

#### 2.4.4. Electrical Resistance

A four-compartment cell was used to determine the area resistance of the nanocomposite membrane. The tested membrane with an effective area of 3.14 cm^2^ was first immersed in a 0.5 M NaCl solution for one day. The 0.5 M NaCl was placed in one compartment of the two-cell compartment on sides of the membrane. The two remaining electrode compartments (one on each side) were filled with a 0.5 M Na_2_SO_4_ solution. Two peristaltic pumps were connected to one specific feed solution compartment and then to an assigned electrode compartment, respectively. A direct current power supply varying from 0.02 A to 0.32 A was provided by DC power. After the system ran steadily, a UT61E digital multimeter was used to connect two CHI111 Ag/AgCl reference electrodes (Fisher Scientific, Fairlawn NJ, USA) to detect the potential differences in the membrane. After measuring the I/V curve, resistance was obtained. The membrane resistance was the value of the obtained resistance subtracting the blank resistance without a membrane.

### 2.5. Power Generation of the Nanocomposite Membrane

The power generation of the nanocomposite membrane was tested in a laboratory-scale RED stack, which consisted of three pairs of CEM and AEM (CJMA-3, thickness 0.15 mm (±0.2 mm), area resistance 4.0 Ω.cm^2^ (±0.5), Hefei Chemjoy Polymer Materials Co., Ltd. Hefei, China) stacked alternately, and two titanium mesh end electrodes coated with iridium plasma. A pair of commercial CEMs (CJMC-3, thickness 190 μm (±0.2 mm), area resistance 3.0 Ω.cm^2^ (±0.5), Hefei Chemjoy Polymer Materials Co., Ltd. Hefei, China) were used as shielding membranes at the two ends of the stack (Figure 1). The electrode washing solution was NaCl (0.25 M) with K_4_Fe(CN)_6_ (0.05 M) and K_3_Fe(CN)_6_ (0.05 M), which was input by a peristaltic pump. The membranes were separated by woven fabric spacers (thickness 0.73 mm, Zhejiang Hailante Protection Materials Co., Ltd., Hangzhou, China) to form small compartments. NaCl solutions with a concentration of 0.5 M and 0.017 M were adopted to simulate the seawater and river, respectively, which were fed at a constant flow rate of 1.3 cm·s^−1^ alternately into the compartments through peristaltic pumps. The effective area of CEM was 12 cm × 5 cm. The RED membrane performance stack was investigated by a four-electrode configuration electrochemical workstation with an external potentiostat in a galvanostatic mode. At each measurement, 10 current steps of 5 s each were applied. The gross power density could be harvested by the maximum value of voltage times current, dividing the total effective area of the tested membranes.

## 3. Results and Discussion

### 3.1. MOF-Derived Nanoporous Carbon Characterization

The SEM and TEM images of ZIF-8 and its derived carbon samples are shown in Figure 2. ZIF-8 possesses a typical cubic hexahedron structure. The average particle size is about 400 nm (Figure 2a). HZIF-8 maintained a similar structure to ZIF-8 (Figure 2b), but an obvious hollow structure could be identified from the broken cubic particles in the image due to the etching process. After carbonization, HZCs also retained similar morphologies to HZIF-8 (Figure 2c). This suggests that carbonization did not change the morphology of MOFs.

The TEM image further reveals the solid cubic structure of ZIF-8 (Figure 2d). The hollow structure can also be observed from the TEM image of HZIF-8 in Figure 2e, which indicates that the TA etching process could successfully transform a solid structure of ZIF-8 into a hollow structure. During the etching process, free protons from TA could break the framework of ZIF-8 and penetrate into it. Furthermore, because of the coating on the surface of ZIF-8, TA also could protect ZIF-8 from further etching [23]. After carbonization, HZCs retained a similar morphology to HZIF-8 with an average particle size about 300 nm (Figure 2f). 

The XRD patterns of samples are presented in Figure 3. The positions of the diffraction peaks, i.e., the as-synthesized HZIF-8, were consistent with the simulated ZIF-8, suggesting that HZIF-8 possessed the same crystal structures as ZIF-8. The results indicated that the chemical etching progress transformed the solid structure into a hollow structure, but had no effect on its crystal structure. The characteristic peaks of HZCs appeared at ~25° and ~44°, corresponding to (002) and (101) crystal planes of the graphitic structure, respectively.

### 3.2. Membrane Characterization

FTIR spectra of nanocomposite CEMs with different amounts of HZC, from 0 wt.% to 1.5 wt.%, are shown in Figure 4. All membranes had a C-H stretching vibration at 2868 cm^−1^ and 2970 cm^−1^. Meanwhile, the C-O stretching vibration also could be confirmed at 942 cm^−1^. These peaks represent the blank PPO polymer. The characteristic absorption peak at 1060 cm^−1^ represented the –SO_3_H, derived from the sulfonation reaction where –SO_3_H was replaced by the aromatic rings of PPO [12]. An obvious peak between 3300 and 3500 cm^−1^ could also be observed, corresponding to the hydrogen reaction between –OH groups and –SO_3_H [2]. The above results indicate that PPO polymers were functionalized by –SO_3_H. However, for HZCs, most groups of its precursor (ZIF-8) disappeared during carbonization at high temperatures. Therefore, no obviously different peaks appeared between the sPPO membrane and the membrane with HZCs. 

Figure 5 shows the thermal stability of the membranes before and after HZCs loading examined with a thermogravimetric analyzer. All of the membranes loss less than 15% in weight up to 400 °C, confirming that this type of sPPO-based membrane confers a high thermal stability. Furthermore, HZCs fillers did not dramatically change the decomposition temperature of the composite membranes, indicating that the addition of HZCs fillers might neither alter nor be involved in the thermal degradation reactions of the polymer.

The mechanical properties of the membranes at different HZCs loadings are shown in Figure 6.

The tensile strength (TS) and the elongation at break (Eb) were significantly different with the increase in loading of the nanoparticles. As the loading continued to increase from 1.0 wt.% to 1.5 wt.%, the mechanical properties declined, which may be attributed to the separated morphology between sPPO and nano particles under higher loadings, confirmed by the latter SEM images. In real applications, the content of incorporated HZCs should be tuned in order to obtain both high tensile strength and moderate elongation at break to ensure flexibility. From this point view, 0.7 wt.% loading was supposed to be an optimum loading of nanoparticles. However, from the later physicochemical and electrochemical test of the nanocomposite membranes, it was found that best properties of the nanocomposite membranes all appeared at a loading of 1.0 wt.%. Therefore, an optimum loading of 1.0 wt.% HZCs was employed to the produced membranes, considering the desirable mechanical properties at the same time.

Figure 7 presents the SEM of the cross-sectional structure of the membranes. Membranes with different amounts of additional HZCs had compact structures without obvious gaps, indicating that the interface between the polymer matrix and the filler was well compatible. The cross section of the blank sPPO membrane was very smooth. As the HZC loading increased, the cross section of the membrane became more and more rough. This phenomenon suggested that the addition of HZCs could change the inner microstructure of the nanocomposite membrane.

### 3.3. Physicochemical and Electrochemical Properties of the Membrane

The physicochemical and electrochemical properties of CEMs under different loadings of HZCs were evaluated in terms of IEC, SD, permselectivity, and area resistance, as shown in Figure 8.

The IEC of the membrane increased as the loading of HZCs increased to 1.0 wt.%, while it decreased as the loading further increased to 1.3 wt.% and 1.5 wt.%, respectively. These results could be attributed to the reconstruction of the microstructure of the membrane by the incorporation of HZCs. Before the incorporation of nanoparticles, some dead channels may exist in the polymer matrix, in which ions could not be transported through. After loading an appropriate amount of HZCs (1.0 wt.% in this case), the dead channels could be opened by the nanoparticles, which made it easier for ion transportation and increased the number of effective ion-exchange functional groups. Moreover, there could be another case that some nanoparticles could rebuild new ion transport channels by interconnections inside the membrane in order to format continuous inner networks. However, many more nanoparticles could aggregate inside the membrane matrix, which would block the open channels and disable some –SO_3_H groups. Therefore, with the further increase in the loading of HZCs, the membrane’s IEC decreased remarkably.

Swelling degree (SD) can affect the permselectivity and area resistance of the membrane. In Figure 8, the SD of the nanocomposite membrane was kept at a low value around the loading of 0.7 to 1.0 wt.%. However, when the loading was less than 0.7 wt.% and more than 1.0 wt.%, the SD increased. Normally, SD increases with the increase of IEC and decrease of the cross-linking degree of membrane [24]. Therefore, as the loading was less than 0.7 wt.%, the increase of SD resulted from the higher IEC of the membrane, inducing more water absorption. When the loading rose from 0.7 to 1.0 wt.%, it was possible for nanoparticles to be cross linked with the sulfonate groups in the sPPO matrices, resulting in a decrease of SD [14]. When the loading of nanoparticles exceeded 1.3 wt.%, SD increased again, due to the aggregation of many more HZCs, whose mesopores could hold too much water in the membrane. 

As shown in Figure 9, the permselectivity of membrane increased with the increase of HZC loading from 0.2 to 1.0 wt.%, but the permselectivity decreased as the loading increased (e.g., to 1.3 wt.% and 1.5 wt.%), achieving the highest value at 1.0 wt.%. Area resistance underwent a reverse phenomenon when it received the lowest value when the weight exceeded 1.0 wt.%, and a higher value when the loading was 1.0 wt.% or less (e.g., 0.2 to 0.7 wt.%). Before the loading reached 1.0 wt.%, the increase in permselectivity was synchronized with the increase in the membrane IEC. When the membrane had a higher IEC, more effective functional groups and ion transport channels appeared in the membrane, which favored ion transport and resulted in the decrease in area resistance. However, when the loading exceeded 1.0 wt.%, the aggregated nanoparticles could block the ionic channels and destroy the membrane inner structures, leading to a decrease in the number of effective ion exchange sites. Thus, the area resistance increased at higher loadings. 

### 3.4. Power Generation Evaluation

Figure 10 presents the gross power density obtained from the nanocomposite membrane at the different HZC loadings. The gross power density of the nanocomposite membrane increased with the increase of HZCs loading up to 1.0 wt.%, and then decreased with the further increase loading at 1.3 and 1.5 wt.%. The highest gross power density was 0.45 W/m^2^, appearing at the loading of 1.0 wt.%, which corresponded to the best physicochemical and electrochemical properties of the nanocomposite membranes obtained previously in the same loading of HZCs. 

As it is known, the power density of the RED system is affected by factors such as the number of cells, flow rate, and the thickness of spacers, besides the membrane properties [14,25]. Even if the stack uses the same IEMs, the harvested gross power densities are different. Therefore, it is difficult to compare the output of the different RED systems. However, the power generation properties of different IEMs can be compared in the same stack and the same working conditions. Furthermore, the improved value ratios of the power density of different IEMs in the different stacks can also be compared before and after nanomaterial incorporation. In this study, the highest power density was 87.5% (about 1.87 times) greater than that of the blank sPPO membrane (0.24 W/m^2^). Compared with that of the membranes incorporated with other nanomaterials reported in the literature (Table 1), obviously, the improved ratio of the power density before and after HZC incorporation was the highest. Therefore, MOF-derived nanoporous carbon nanomaterials can act as promising fillers to improve the performance of IEMs in power generation.

## 4. Conclusions

In this study, MOF-derived nanoporous carbon nanoparticles were successfully incorporated into an sPPO polymer in order to improve the performance of the nanocomposite CEM. The incorporated HZCs tailored the inner structure of the membrane matrix, affecting the physicochemical and electrochemical properties and power generation ability of CEM. An appropriate loading of nanoparticles facilitated the microstructure of the membrane, providing the synthesized membrane with a super IEC, improved permselectivity, the lowest SD, and area resistance. The highest gross power density of the nanocomposite membrane was up to 0.45 W/m^2^, which was 87.5% greater (about 1.87 times) than that of the blank sPPO membrane at a loading of 1.0 wt.% HZCs. Therefore, the incorporation of MOF-derived nanoporous carbon nanomaterials into polymer membranes is a promising approach to enhance the performance of the RED system, and such newly-developed nanocomposite membranes have great potential for energy generation.

## Figures and Tables

**Figure 1 membranes-12-00322-f001:**
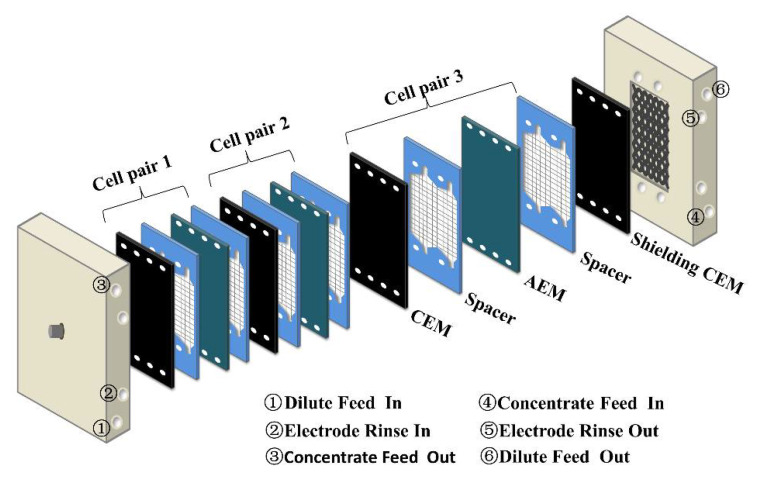
The RED stack used in the experiment.

**Figure 2 membranes-12-00322-f002:**
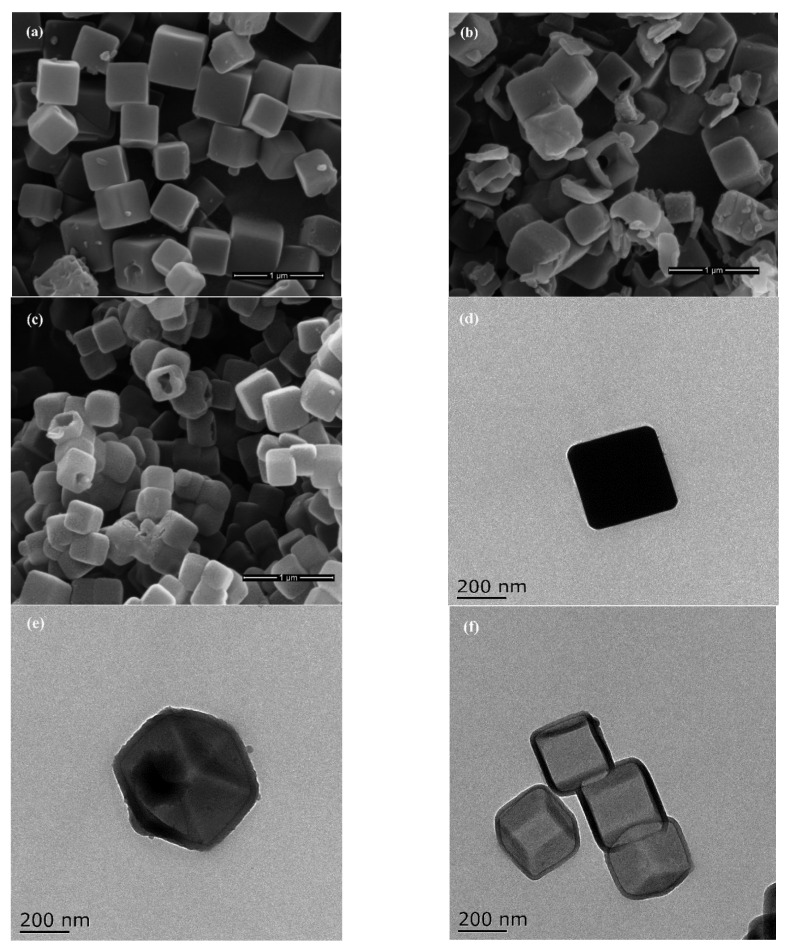
SEM images of (**a**) ZIF-8, (**b**) HZIF-8 and (**c**) HZCs in the top row followed by.TEM images of (**d**) ZIF-8, (**e**) HZIF-8 and (**f**) HZCs.

**Figure 3 membranes-12-00322-f003:**
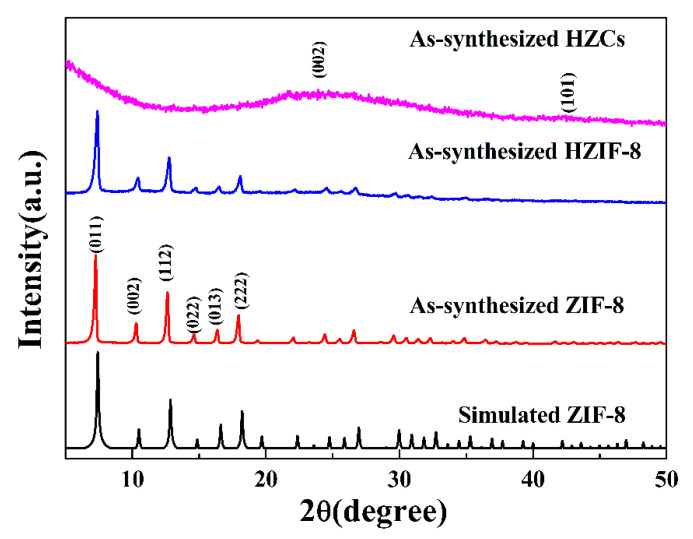
XRD pattern of ZIF-8, HZIF-8 and HZCs.

**Figure 4 membranes-12-00322-f004:**
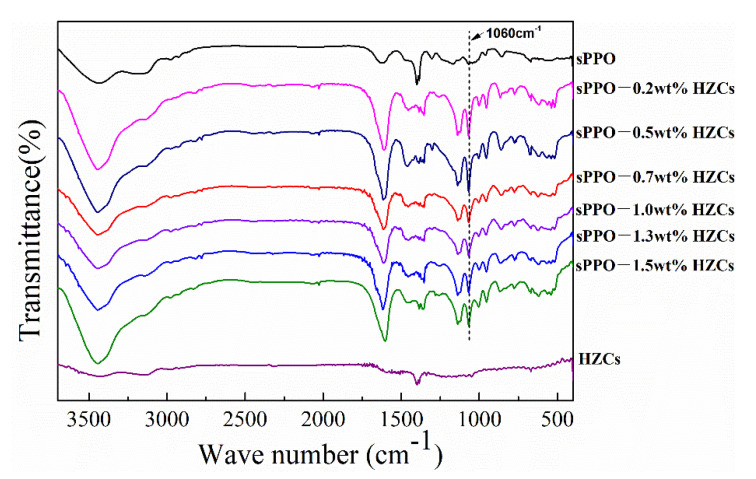
FTIR spectra of nanocomposite membranes.

**Figure 5 membranes-12-00322-f005:**
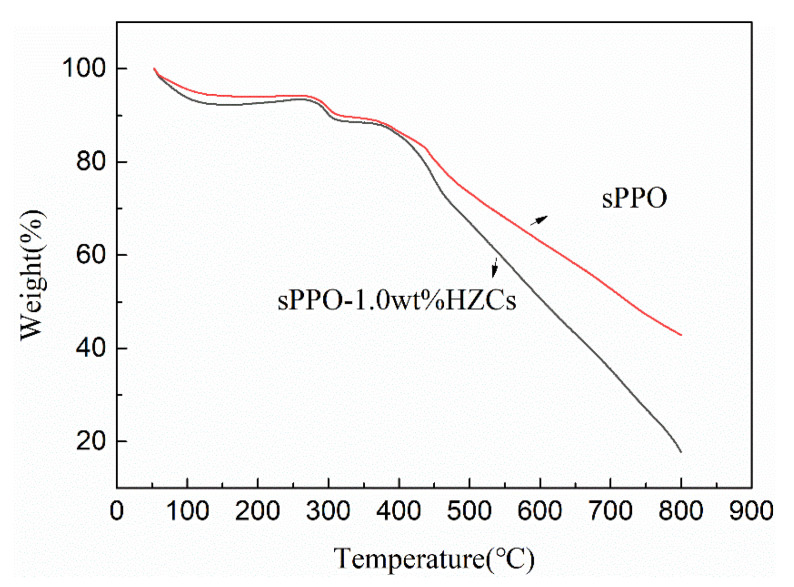
TGA of the membranes before and after HZCs loading (1.0 wt.% HZCs).

**Figure 6 membranes-12-00322-f006:**
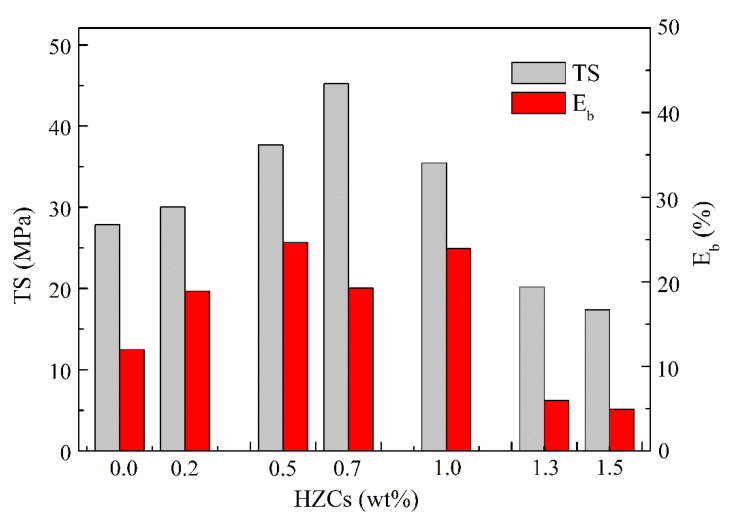
Mechanical strength of the nanocomposite membranes.

**Figure 7 membranes-12-00322-f007:**
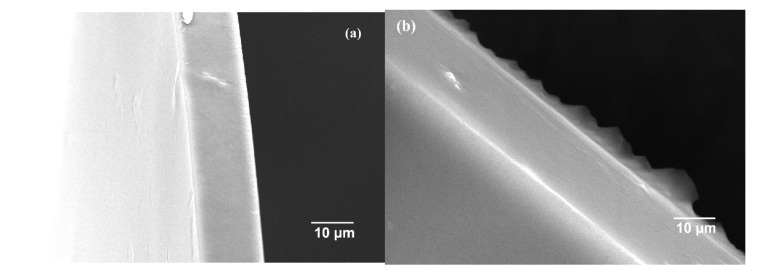
SEM of cross section surface of CEMs. (**a**) Blank sPPO membrane; (**b**) 0.2 wt.% HZCs-sPPO; (**c**) 0.5 wt.% HZCs-sPPO; (**d**) 0.7 wt.% HZCs-sPPO; (**e**) 1.0 wt.% HZCs-sPPO; (**f**) 1.3 wt.% HZCs-sPPO; (**g**) 1.5 wt.% HZCs-sPPO.

**Figure 8 membranes-12-00322-f008:**
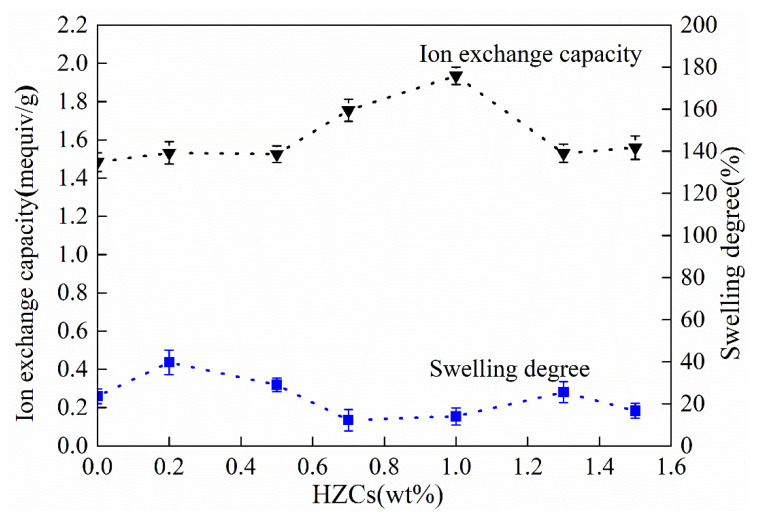
Effect of HZCs loading on IEC and SD of the nanocomposite membrane.

**Figure 9 membranes-12-00322-f009:**
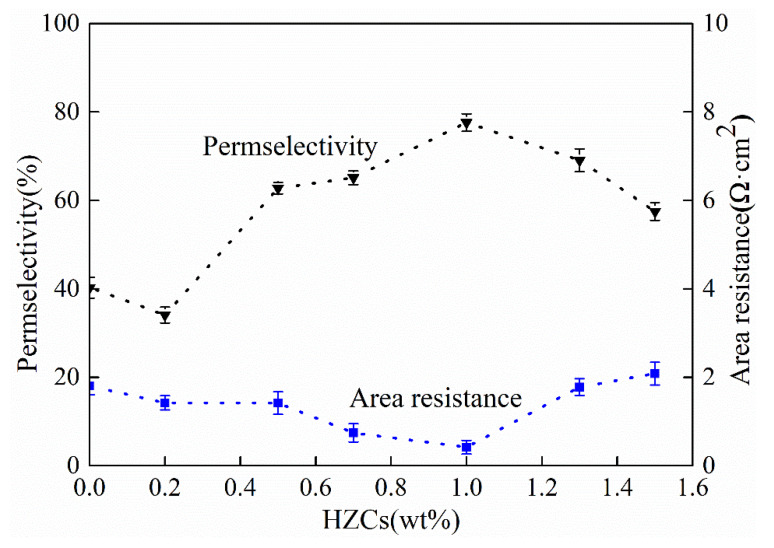
Effect of HZCs loading on the permselectivity and area resistance of the nanocomposite membrane.

**Figure 10 membranes-12-00322-f010:**
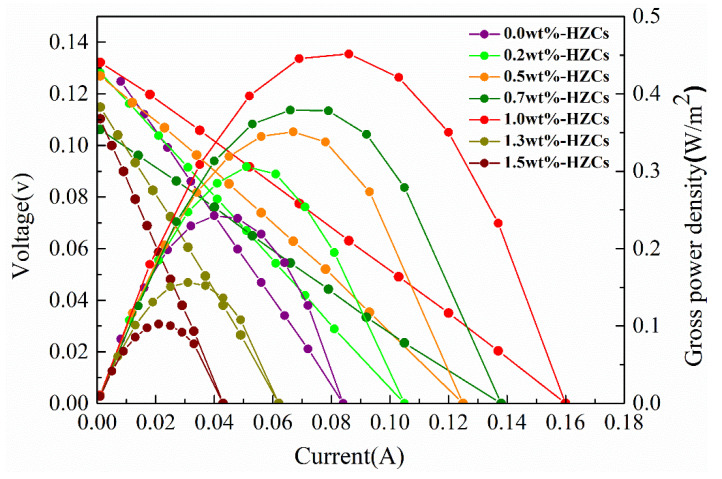
Power density curves obtained from the nanocomposite membrane at different HZC loadings.

**Table 1 membranes-12-00322-t001:** Properties and power density of the nanocomposite membranes in a RED system.

Nanomaterials	The Blank Membrane	*^a^* The Physicochemical Properties of Nanocomposite Membranes				Gross Power Density of the Blank Membrane (W/m^2^)	Maximum Gross Power Density of the Nanocomposite Membrane (W/m^2^)	Improved Ratio of Power Density (%)	Refs.
		Permselectivity (%)	IEC (meq/g)	SD (%)	Area Resistance (Ω/m^2^)				
0.7 wt.%-Fe_2_O_3_–SO_4_^2−^	sPPO	87.65	1.40	26.00	0.97	0.98	1.3	32.65	[2]
PDDA	PVA	59	1.48	17.50	0.77	*^b^* 0.34	*^c^* 0.58	70.59	[7]
5.0 wt.%-sPVA	sPPO	87	1.98	48	1.55	0.41	0.46	10.87	[11]
*^d^* 0.5 wt.%-SiO_2_–SO_3_H	sPPO	81.40	0.99	33	0.95	1.08	1.3	20.37	[12]
0.5 wt.%-O-MWCNT	sPPO	95.2	2.27	41	0.5	0.36	0.48	33.33	[14]
1.0 wt.%-HZCs	sPPO	77.61	1.94	14.01	0.42	0.24	0.45	87.50	This study

*^a^* The optimum values of the membrane in the literatures. *^b^* The value was obtained at blend ratios (m (PDDA)/m(PVA)) of 0.75. *^c^* The value was obtained at blend ratios (m (PDDA)/m(PVA)) of 1.5. *^d^* a particle size of 70 nm.

## Data Availability

Not applicable.

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
