# Peer review of "MOF-Derived Nanoporous Carbon Incorporated in the Cation Exchange Membrane for Gradient Power Generation"

_membranes, 2022, doi:10.3390/membranes12030322_

Round 1
Reviewer 1 Report
Comments
The present manuscript demonstrates a nanocomposite cation exchange membrane for reverse electrodialysis application. Nanocomposite membrane comprises poly (2,6-dimethyl-1,4-phenylene oxide) (PPO) as a polymer matrix and MOF-based nanoporous carbon as filler. SEM, TEM, XRD, and FTIR spectroscopy were used for characterizing the prepared material and membrane. Moreover, the nanocomposite membrane's physicochemical properties, such as: IEC, swelling density, perm-selectivity, and bulk resistance, are also reported in the manuscript. The RED cell was composed of 3 cell pairs containing electrodes, the prepared nanocomposite membrane as CEM, with CJMA-3 as an AEM. Plus, feed solution containing NaCl-based high and low concentration salt solution.
This manuscript is unable to reveal the material potential. The present manuscript addresses a routine work that does not contain novelty. Several similar systems have been reported with better physicochemical and cell performance. Therefore, I cannot recommend this article for publication in the present form. Some observations are listed below:
1. Page 2, line 57 and page 11, line 303 mentioned that the mechanical strength of the membrane is an essential parameter for developing a high-performance device. Still, there is no result presented in the main text. It is recommended that the author include the prepared membrane's mechanical strength.
2. What is the primary motive for choosing PPO as a polymer matrix? Why not other high-performance polymer matrices such as sPEEK or Nafion? Please justify, be thoughtful.
3. Introduction section requires an intense literature survey on nanocomposite membranes in terms of reverse electrodialysis device cell performance.
4. A fundamental question arises here, what is the main reason for choosing organic-inorganic as a filler? As the author mentioned on page 2 (an increased porosity ………….RED system). Why not only organic fillers such as carbon-based material, other organic frameworks, etc., which are also known for enhancing membrane performance.?
5. Author must provide the comparison table that must contain physicochemical properties (such as perm-selectivity, ion-exchange capacity, mechanical strength, etc.) of the prepared membrane and the conventional membranes.
6. On the other hand, a detailed study must be provided to validate the obtained values.
7. It also lacks the thermal properties of the prepared membrane. It should be included in the main text.
8. Figure 8 must be supported with the voltage and power density curve and represents the equation used for calculation.
9. What is the device's energy conversion efficiency, and please demonstrate the hydrodynamic loss of the cell? For a better understanding.
10. It is recommended that the author expand table 1 by showing more data.
By categorizing the organic-inorganic filers, inorganic fillers, and organic fillers separately.
So that more readers are attracted to the manuscript.
Note: It is advised that all authors must present their manuscript more strongly to the reader. Detail explanation is required in the result and discussion section. The author must perform an intense literature survey and be encouraged to include more relevant references in the manuscript.
Reviewer 2 Report
The work under review proposes a self-made cation-exchange membrane with embedded nanoparticles in the RED membrane stack. The main problem with the manuscript is the lack of data on the RED performance as in the methodology section and the results section.
There are some concerns about the quality of the work that needs to be addressed first:
Line 35, “power cubic meter” should be “power per cubic meter”. Also, please check the overall language in the whole manuscript.
Line 46 “with ions containing polymer electrolytes” please clarify the sentence.
Lines 51-53 “Therefore, the performance of IEMs has a great impact on the RED system, including surface morphology, thickness, swelling degree (SD), ion exchange capacity (IEC), permselectivity, and resistance of membranes” please rewrite the sentence. Now it says that the RED system poses properties such as surface morphology, which are affected by IEMs.
Section 2.2.1 What is the concentration of the resulting sPPO solution in methanol? Please provide. Section 2.2.3 “19 wt.% sPPO polymer solution was dissolved in DMSO” what is the solution that was dissolved? What is the composition of the resulting solution?
Section 2.2.4 “Membrane resistance was the value of the obtained resistance subtracting the blank resistance without a membrane.” The obtained value is the sum of ohmic resistance of the membrane and non-ohmic components of the diffusion boundary layers resistances. The latter is also included as in the blank experiment without membrane, there is no concentration polarization thus only the ohmic part of the total resistance is subtracted. The measurements must be carried out under conditions when polarizing current is minimal, thus negating the concentration polarization or under alternative current to exclude non-ohmic components. As such, the resistance data provided in the manuscript cannot be compared to the data from other research groups.
Line 210 “coated with iridium plasma” what is iridium plasma, and how does it interact with the solution?
Section 2.5 “The RED membrane performance stack was investigated by a four-electrode arrangement in a galvanostatic mode. The gross power density could be harvested by the maximum value of voltage time’s current dividing the total effective area of the tested membranes” The method for measuring the power characteristic requires a better explanation. Not it looks like the stack was polarized by an external current source, i.e., it performs as an ED cell.
Section 3.4 What was the external resistance value? The OCV value? Provided data is not enough to compare it with other sources. Please provide the power density vs. load resistance curve.
Reviewer 3 Report
The present article describes the novel membrane with incorporated nanoparticles and reports several properties of these membranes. The pattern of change of properties with increasing load of modifier reminds me of the hypothesis of the limited elasticity of the membrane pores, as in https://doi.org/10.1016/j.mencom.2010.05.011 (note that I'm not insisting you should cite that, just suggesting a short article you might look through). All in all the manuscript is logically written and it leaves a good impression, but sometimes I'm unsure of English language (see "to simulate the seawater and river", lines 217-218), so the authors might consider a reread. The description of the experiments might be more detailed, for example, section 2.4.3 might provide a time point at which the potential drops were registered and of the time drops were kept constant between the different experiments.
There are also several small typos that I'm sure would be fixed at proofreading (such as missing spaces, as in "Thickness 0.73mm" at line 215).
Round 2
Reviewer 1 Report
The author of this manuscript dodges most of the reviewer's questions. Instead of providing proper justification and evidence to the reviewer's questions, the authors overconfidently ignore the importance of the reviewer's constructive feedback.
Disregarded answers are as follows:
Ans 1 Next study? I wish the author's subsequent study would be a breakthrough in membrane technology—best wishes for that. But Ans 1 still requires data, not their proposed work.
Ans 3. Where are those literature survey articles? Please incorporate this in the current study.
Ans 4 Author misunderstood my question.
In a simple word: Where is N2 absorption DATA?
Ans 5. Authors feel that presenting the comparative physicochemical properties of the membrane is not necessary.
Suppose I accept the author's answer. Thus, there is no need for membrane physicochemical parameters. Then, why do the author-present IEC, swelling density, and perm-selectivity of their membrane? Just obtain the power density value and done.
Ans 6. Next study?
Ans 9. The authors ignore the hydrodynamic loss question.
Ans 10. The author again tries to neglect the question. Instead of providing data, the author prefers to advise back to reviewers.
Overall, reviews get the impression of the author's laziness. Authors are encouraged to provide thoughtful answers. If necessary, use the supporting information section.
Reviewer 2 Report
The authors have taken into account all of my comments. I thereby recommend the manuscript to be published in its current form
Author Response
We have revised the manuscript again. Thanks for your approval.
Round 3
Reviewer 1 Report
I recommend this article for the further publication process.